# Topology-Optimization-Based Learning: A Powerful Teaching and Learning Framework under the Prism of the CDIO Approach

**Evangelos Tyflopoulos *** , **Cecilia Haskins** and **Martin Steinert**

Department of Mechanical and Industrial Engineering, Norwegian University of Science and Technology (NTNU), 7491 Trondheim, Norway; cecilia.haskins@ntnu.no (C.H.); martin.steinert@ntnu.no (M.S.)
* Correspondence: evangelos.tyflopoulos@ntnu.no; Tel.: +47-7341-262-3

**Abstract:** Topology optimization (TO) has been a useful engineering tool over the last decades. The benefits of this optimization method are several, such as the material and cost savings, the design inspiration, and the robustness of the final products. In addition, there are educational benefits. TO is a combination of mathematics, design, statics, and the finite element method (FEM); thus, it can provide an integrative multi-disciplinary knowledge foundation to undergraduate students in engineering. This paper is focused on the educational contributions from TO and identifies effective teaching methods, tools, and exercises that can be used for teaching. The result of this research is the development of an educational framework about TO based on the CDIO (Conceive, Design, Implement, and Operate) Syllabus for CAD engineering studies at universities. TO could be easily adapted for CAD designers in every academic year as an individual course or a module of related engineering courses. Lecturers interested in the introduction of TO to their courses, as well as engineers and students interested in TO in general, could use the findings of this paper.

**Keywords:** topology optimization; education; teaching methods; CDIO

## 1. Introduction to Topology Optimization (TO)

Topology optimization (TO) is one of the most commonly implemented optimization categories in structural optimization (SO) [1,2]. The design domain of a structure is discretized, and then unnecessary material is either removed or moved to create a layout that meets the given objective functions and constraints of the structure. TO is mainly used by engineers who are interested in material reduction or other optimization objectives, such as stress, deflection, and cost.

Bendsøe and Kikuchi [3] developed the homogenization method in order to solve the topology optimization problem. According to the homogenization theory, the design domain of a structure is discretized into unit cells. These microstructures are used in the calculation of global material properties. Since 1988, several gradient-based and non-gradient-based techniques have been developed [4]. On the one hand, the solid isotropic material with penalization (SIMP) method [5], as well as the evolutionary structural optimization (ESO) method [6], are two notable examples of gradient-based techniques. On the other hand, the application of genetic algorithms [7] that explore a whole population for possible solutions in TO is worth mentioning as a non-gradient-based technique.

TO has been applied on the macro-, meso-, and micro-scale levels. In addition, several methods have been developed for the implementation of multi-scale TO [8,9]. Hence, there is a wide range of TO applications in the industry—from large, complex structures, such as airplanes, to small antennas, micro-machines, fluids, dynamics, multi-physics, and customized human implants. Furthermore, TO has been adapted in both architecture and art for design inspiration. The Qatar National Convention Center [10] and the TO

of furniture [11] are two notable examples in the two latter categories, respectively. TO, at its current state of the art, is mostly used as a design process. Usually, topologically optimized designs must be redesigned in order to be manufactured with conventional production methods (CPMs). In recent years, much research has been committed to additive-manufacturing-oriented TO, wherein the derived TO design solutions can be produced directly [12]. Figure 1 illustrates that the implementation of TO is a combination of classical mechanics, mathematics, computer programming, finite element methods (FEMs), computer-aided design (CAD), 3D printing, and conventional production methods (CPMs). The inclusion of so many methods suggests that TO can also be utilized as a source of computational exercises across a broad range of engineering curricula.

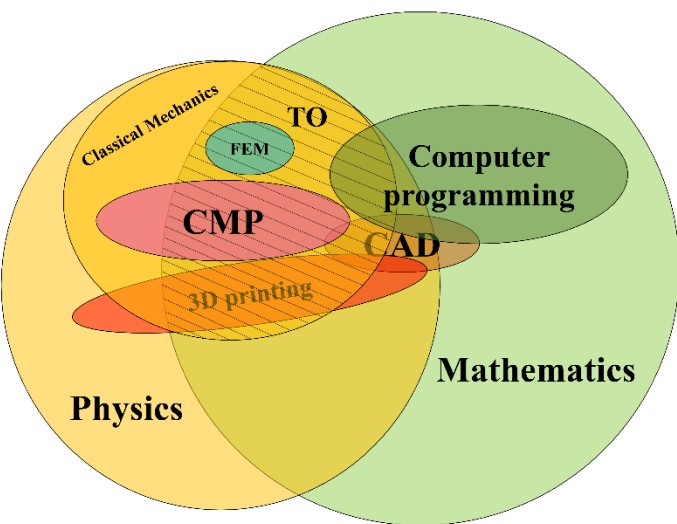

**Figure 1.** A Venn diagram of TO as a multi-educational tool.

In this paper, the authors focused on the educational aspects of TO. Before engineers in industry can see TO as a useful tool for material and cost savings as well as design inspiration, the authors suggest educating the next generation of CAD designers by using TO as a valuable teaching tool that can provide multi-disciplinary knowledge to undergraduate students. The recommendation is to introduce TO in an easily taught way to students studying CAD engineering from their first academic year by providing topology-optimization-based learning (TOBL) under the prism of the CDIO (Conceive, Design, Implement, and Operate) Syllabus. TOBL serves as a knowledge bridge to the essential elements of a CAD design degree program, such as 3D modeling and finite element methods (FEMs), by using the TOBL framework that was developed as the main result of this research. To demonstrate the utility of TO in CAD design education, the authors examined the following questions:

- Can TOBL be used effectively in a degree for CAD design?
- How easy is it to introduce elementary TO to the under- and postgraduate students, and is there an effective teaching method?
- What is the prerequisite knowledge that is needed to teach the fundamental TO theory?
- At which level can TO be introduced? Are there differences between teaching TO to undergraduate and postgraduate students?

The structure of this paper is as follows: In Section 2, the CDIO approach is presented. Then, the general structural optimization problem is described in Section 3, leading to Section 4, which contains examples of tools, software, games, and exercises that could be effectively used as active learning tools in the teaching of TO. The findings from the previous sections constitute the background of the development of an educational framework for TO in Section 5. The developed framework is discussed in Section 6. Finally, Section 7 concludes with the valuable contributions of this research.

## 2. CDIO: An Effective Educational Framework

The changing needs in modern engineering have motivated academics to reconsider engineering education. The industry expects newly graduated engineers to possess the basics, to bring new skills with them into the workplace, to apply knowledge of mathematics and engineering, to design new products and processes, to communicate effectively, to function in multi-disciplinary teams, and to use new techniques and modern tools [13]. Since the 1960s, there has been a return to the roots in engineering education, i.e., from theoretical to practical engineering. The participation of the students in the learning process is increasing gradually with the introduction of in-class active learning tools. Active learning is a teaching method where the students have a central role by learning through games, activities, and crafts, as well as by communicating and working in groups and projects [14]. Different pedagogical methods have been developed in the last decades, including work-based learning (WBL) [15], practice-based professional learning (PBPL) [16], problem-/project-based learning (PBL) [17], and design-based learning (DBL) [18], to mention a few. All of these methods have the same denominator, which is increasing students' knowledge acquisition, comprehension, and intuition, as well as stimulating their motivation to learn by using theory in practice in the classroom. This pedagogical strategy aligns the theory with the practical implementation, so the students can learn both the theory's applicability and its limitations.

In the late 1990s, an innovative educational framework for engineers was developed based on this strategy under the name of CDIO [19]. The name of the framework is an acronym of the major learning phases of Conceiving, Designing, Implementing, and Operating. CDIO began as an initiative by four universities: Massachusetts Institute of Technology (MIT), Chalmers University of Technology, KTH Royal Institute of Technology, and Linköping University. Their intention was to present a of the UNESCO's universal educational taxonomy developed in 1996 that was extended and focused on engineering [20]. The CDIO approach describes all of the activities that are needed during the total lifecycle of a product, process, or system. The first activity is to identify the stakeholders' needs and strategies and to create project and business plans. This activity is described by the word "conceive". The word "design" is the second activity, which is the creation of any type of design (plans, drawings, algorithms, etc.) that describes the product or the process that will be implemented. The third activity, "implement", is the transformation of the design concepts into the product or process, as well as testing and validation of how well they perform. Finally, "operate" is the last activity, which is the use of the product or process for its intended purpose, as well as its maintenance, evolution, recycling, and retirement [17]. The implementation of the CDIO framework is described in detail in the CDIO Syllabus report [21]. In addition, the framework was designed as a template with instructions for adoption in any engineering education institution. The CDIO community has grown to include approximately 120 university members worldwide since its inception.

### 2.1. The CDIO Syllabus and Its Standards

The CDIO Syllabus was developed based on feedback from academics, industries, under- and postgraduate students, and practicing engineers. It thoroughly describes the full set of knowledge, skills, and attitudes that a modern engineer should possess after his/her graduation and their level of proficiency. The revised version (second version) of the CDIO Syllabus, at its first level, consists of four main parts: 1. Disciplinary knowledge and reasoning, 2. Personal and professional skills and attributes, 3. Interpersonal skills, teamwork, and communication, and 4. Conceiving, Designing, Implement, and Operating systems in the enterprise, societal, and environmental contexts. The CDIO Syllabus is described in "The CDIO Syllabus v2.0: An Updated Statement of Goals for Engineering Education" at the first, second, third, and fourth levels [21]. However, at its second level of detail, the CDIO Syllabus is considered sufficient for a course or module design.

The first section of the Syllabus is the UNESCO's "Learning to know", and it describes the expected knowledge, such as mathematics, physics, and engineering fundamentals,

that the students should gain from their study program. The content of this section can vary among the different study programs based on their particular needs. The remaining three sections include the knowledge, skills, and attributes that are required from all engineering graduates regardless of their specialization. Specifically, the second section is about the personal learning outcomes of the students, ranging from problem solving, experimentation, and system thinking to attitudes and ethics. This section is equivalent to the UNESCO's "Learning to be". The interpersonal skills are the focus of the third section, where the students learn to work and communicate in groups. The teamwork and communication presented in this section are very close to the "Learning to live together" described by UNESCO. Finally, UNESCO's "Learning to do" is the conceiving, designing, implementing, and operating (CDIO) described in the fourth and last section of the Syllabus.

The CDIO Syllabus is intended to ensure that students will expand their skills through its implementation. According to the vision of CDIO [19], this depends on the structure of the curriculum, the content of the courses, the learning environment, the teaching method, and the way that the learning outcomes are evaluated and interpreted. For this reason, the CDIO initiative developed 12 principles under the name of CDIO Standards to guide any educational engineering program that embraces the CDIO approach. The utilization of these standards will secure, monitor, and evaluate the implementation of CDIO. According to the CDIO Initiative, the standards can be formed into groups with respect to their context. Standard 1 is considered as the foundational principle of the CDIO approach, as it provides a lifecycle context of education. The next three standards (2–4) are related to the development of an integrated curriculum that can support the CDIO Syllabus. Standards 5 and 6 describe how the ideal design and implementation experiences, as well as the students' required workspaces, should be arranged, while Standards 7 and 8 focus on the teaching and learning methods. The development of faculty is presented in Standards 9 and 10, and finally, Standards 11 and 12 deal with the assessment and evaluation of the study program.

### 2.2. Designing a Course Aligned with the CDIO Approach

Despite the fact that the CDIO approach is mainly applied on a program-level scale, it is also applicable on a course or module level [19]. However, the design of a course that is aligned with the CDIO approach is challenging. It is crucial for the person responsible for the course or the designer to plan the course in relation to the integrated CDIO curriculum and not independently. Thus, the development of the course can be seen from both a top-down and a bottom-up perspective. An illustration of a course design procedure that is aligned with the CDIO approach is depicted in Figure 2.

The development of an integrated CDIO curriculum of a study program is implemented together with all of the stakeholders that share an interest in the graduates, such as the faculty, under- and postgraduate students, and the industry. These key stakeholders evaluate and monitor the development process and elaborate on the needs of a contemporary engineer. The goals of the study program are defined based on these needs, and these, in turn, configure the CDIO Syllabus. At this point, the program leaders and the responsible faculty design the integrated CDIO curriculum by going through all of the included courses and are always in dialogue with the program stakeholders. According to Standard 3, the disciplinary courses should mutually support the curriculum. In addition, personal, interpersonal, and building skills should be a focal point.

The planning of each course begins with the identification of the purpose of the course by defining its learning outcomes. According to Biggs [22], there is a constructive alignment of a course's learning outcomes with its teaching and learning activities, as well as its assessment. The learning outcomes are the knowledge, skills, and attributes that students attending this course are expected to gain. These should be specific, detailed, and realistic with regard to the course's time and resources. Moreover, the learning outcomes can be classified based on the desired level of understanding from the students' side. The CDIO initiative recommends the utilization of the Feiser–Shmitz taxonomy [23]

for the categorization of a course's learnings outcomes. This taxonomy consists of five levels of understanding, including defining, computing, explaining, solving, and judging. It is clear that students have a long path to knowledge from the description and the interpretation of a problem to its solution and evaluation. The teaching and learning activities are all activities that help students to acquire the intended learning outcomes. These activities must support active learning (Standard 8) and, thus, should always embrace practical examples and applications of the presented theoretical concepts. In addition, the students should reflect on their experiences and give constructive feedback. Finally, the assessment is a measure of the extent to which the students have reached the desired learning outcomes. According to Standard 11, the applied assessment methods depend on the course's outcomes. For example, the preferred evaluation of learning outcomes related to design and implementation skills uses the measurement of recorded observations rather than traditional written tests. Examples of such observations could be the delivery of design artifacts or a portfolio of assignment results.

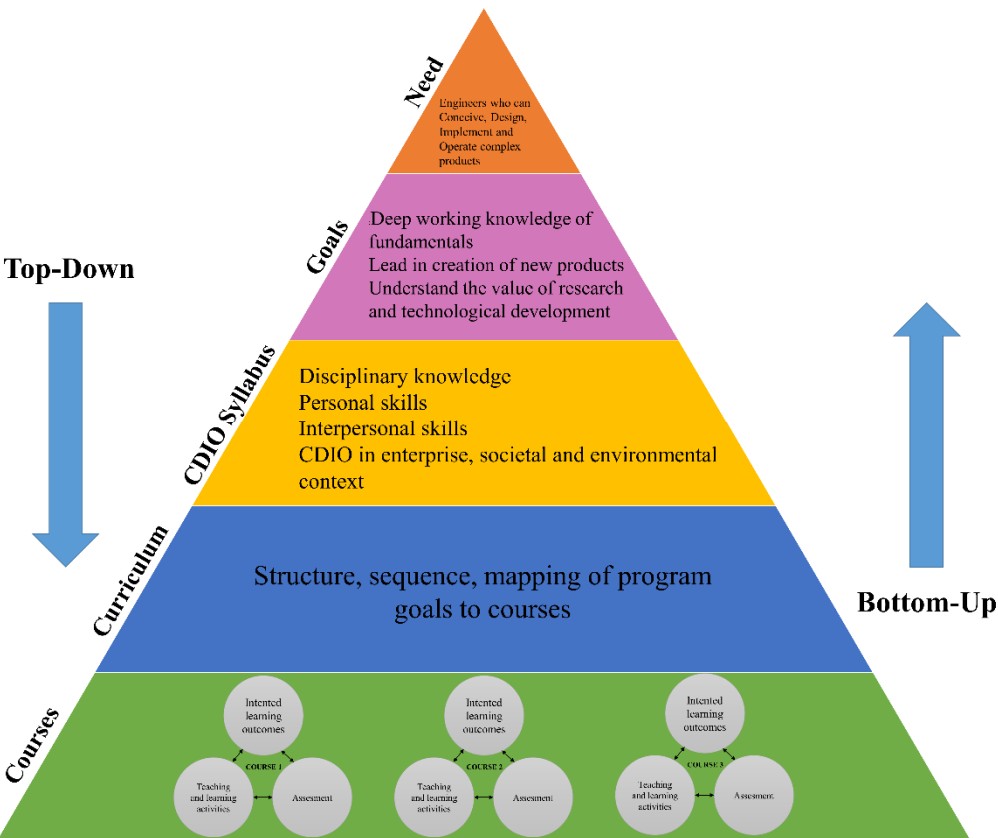

**Figure 2.** The course design procedure based on the CDIO Syllabus and the work of Crawley, Malmqvist, Ostlund, Brodeur and Edstrom [19].

TO can be considered either as an individual course or as a crucial module related to disciplinary engineering courses at the bachelor or master level. A topology-optimization-based learning (TOBL) under the prism of the CDIO approach is presented in this paper. The development of the TOBL will mainly be based on the first section of the CDIO Syllabus at its second level of detail, as well as the section's Standards 1–4. Furthermore, dependencies between the TOBL and an integrated CDIO curriculum will be identified. Open-source ideas and resources provided by the CDIO Initiative assist the rapid adaptation and smooth facilitation of the CDIO approach for any engineering university, including those with limited resources. Therefore, open-source TO tools that can support the TOBL will be presented in this research work. The development of the TOBL and its framework will be

described in Section 5, but first, Section 3 will present the general structural optimization problem, followed by an example of active learning tools in Section 4.

## 3. The General Structural Optimization Problem

Before the development of a TOBL, it is crucial to review the general optimization problem. The general optimization problem is described by the following mathematical formulation [24]:

$$(SO) \begin{cases} minimize/maximize\ f(x,y)\ with\ respect\ to\ x\ and\ y \\ subject\ to \begin{cases} g(y) \leq 0, & behavioral\ constraints\ on\ y \\ g(x) \leq 0, & design\ constraints\ on\ x \\ g(y),\ g(x) = 0, & equilibrium\ constraints. \end{cases} \end{cases} \tag{1}$$

where:

$f(x)$: objective function $f$;
$x$: design variable;
$y$: state variable.

The objective function of a structure can usually measure the cost of production, stress, weight, compliance, and displacement, among other things. The numerical value of this function is used as a criterion for the evaluation of the possible design solutions. In the case of the minimization of an objective function, for example, the minimization of its weight, the lightest design will be chosen as the optimal solution. The design variables ($x$) are either functions or vectors that describe the design and can be changed during the optimization. They represent a characteristic of the design, such as a geometric characteristic or the chosen material. Finally, the state variables ($y$) are either functions or vectors that represent the response of the optimized mechanical structure, such as stress, strain, force, and displacement [24].

Furthermore, the behavioral and design constraints can be combined and written as $g$ ($x,y$). In addition, in a linear discretized problem, the equilibrium constraints are [24]:

$$K(x)u = F(x) \tag{2}$$

where:

$K(x)$: stiffness matrix;
$u$ : the displacement vector;
$F(x)$: the force vector.

Thus, $u = u(x) = K(x)^{-1}F(x)$ can substitute for the state variable $y$ while the equilibrium constraints can be left out from the optimization problem. Hence, the nested formulation of (1) is:

$$(SO)_{nested} \begin{cases} min f(x, u(x)) \\ subject\ to\ g(x, u(x)) \leq 0 \end{cases} \tag{3}$$

The objective function used in traditional TO is the total compliance of the structure's elements. The compliance is the reciprocal of the stiffness, and thus, by minimizing the compliance of the structure, one can increase its robustness. Thus, by formulating the stiffness optimization problem, a density-like variable is assigned to the finite elements created, and thus, $x = \rho$. Hence, Formulation (3) is transformed into [24]:

$$(SO)_{nested} \begin{cases} min f(\rho, u(\rho)) \\ subject\ to\ g(\rho, u(\rho)) \leq 0 \end{cases} \tag{4}$$

In the case of an integer problem, the binary values (0, 1) are used for $\rho$, where 1 means material and 0 is void. A classical method for solving discretized structural optimization problems is the optimality criteria method [24]. However, due to the solutions' complexities and challenges, such as in the checkerboard problem (structural discontinuities), in

optimized design solutions, the discrete values of $\rho$ are replaced with continuous variables, and thus, $0 \leq \rho \leq 1$. In this case, finite elements with intermediate densities are created. Gradient-based algorithms are utilized for the solution of continuous optimization problems. In addition, interpolation methodologies are used for the calculation of the properties of material. The most commonly implemented interpolated method is the SIMP method [24], where the Young modulus of the material is expressed in a continuous setting by using the following power law:

$$E = E_0 + \rho^p (E_1 - E_0) \tag{5}$$

where:

$E$: Young's modulus;
$p$: penalization factor, usually with the values 1–3.

Furthermore, it is very common for there to exist more than one objective function in an optimization problem. For example, a structure could be optimized for both its weight and maximum strength. In other words, the goal of this optimization problem is the identification of the lightest design with the smallest maximum stress. In this case, the optimization problem becomes a multi-objective mathematical problem that can be formulated as [24]:

$$minimize/maximize\ (f_1(x,y),\ f_2(x,y),\ \ldots,\ f_n(x,y)), \tag{6}$$

where $n$ is the number of the objective functions, and the constraints are the same as for (1). It is clear that the different objective functions do not take their max/min values at the same $x$ and $y$. Thus, in order to calculate the optimal solution of (6), Pareto optimality [25] is enforced. This solution, which is also called Pareto optimal, is found for $x = x^*$ and $y = y^*$ and satisfies, in the minimization case, the following constraints [24]:

$$f_i(x,y) \leq f_i(x^*,y^*),\ \text{for all } i = 1,\ \ldots,\ \text{n}, \tag{7}$$

$$f_i(x,y) < f_i(x^*,y^*),\ \text{for at least one } i\ \in\ (1,\ \ldots,\ \text{n}). \tag{8}$$

A transformation of (6) into a scalar objective function contributes to the identification of the Pareto optima by varying the weights in the following formula [24]:

$$\sum_{i=1}^{n} w_i f_i(x,y), \tag{9}$$

where $w_i \geq 0$ for $i = 1,\ \ldots,$ n indicates the weigh factors that satisfy $\sum_{i=1}^{n} w_i = 1$. Interested readers are directed to the works of Bendsøe and Sigmund [2] and Christensen and Klarbring [24] for more analytical calculations.

## 4. Examples of Active Learning Tools in Topology Optimization

TO could be introduced to students by using active learning tools that help students' intuition and comprehension, such as figures, interactive exercises, and games. Figure 3 depicts the initial and topologically optimized designs of a cantilever beam.

Students could be asked to discuss and choose among the proposed design solutions based on their strength, mass, and quality. In addition, they could be divided into small groups that try to build the lightest and strongest optimized cantilever beams by using readily available materials, such as cardboard and MDF. For the building process, the parameters used should be based on the given boundary conditions and constraints, while the groups try to identify the load paths and the critical areas in the beam's structure. All of the developed beams are then checked for their strength and weight, and the best solution is announced. Another example could be the use of TO applications/games, such as 2D and 3D interactive TopOpt apps for handheld devices and web, where the students can

change the load cases and the boundary conditions themselves and can monitor the design solutions and interact with them [26,27], as shown in Figure 4.

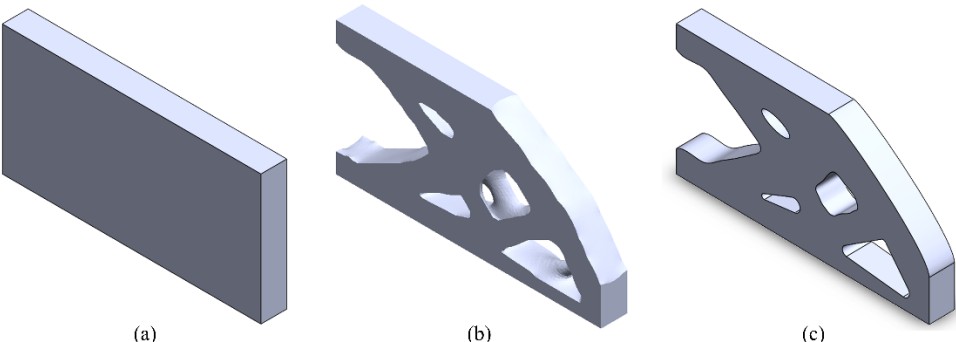

| (a) | (b) | (c) |

**Figure 3.** Topology optimization of a cantilever beam: (**a**) the initial design, (**b**) the TO geometry, and (**c**) the redesigned geometry for CMP.

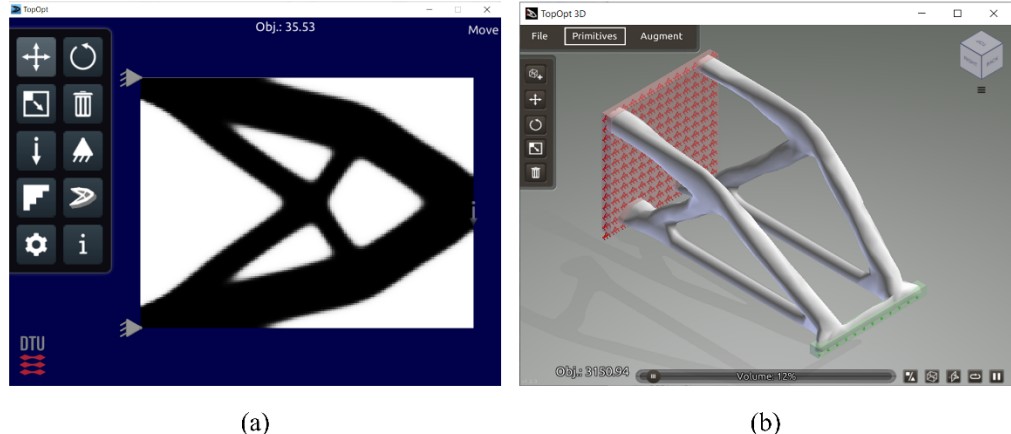

| (a) | (b) |

**Figure 4.** TO games for handheld devices: (**a**) 2D TopOpt app [26] and (**b**) 3D TopOpt app [27].

All of these activities are intended to excite students' curiosity about TO and bring forth an elaboration of how a designer can evaluate the strength of a structure and how he/she can reduce its weight without compromising its strength. From these exercises, the students can understand the advantages afforded by the implementation of TO and, at the same time, they combine their essential knowledge in mathematics, physics, and mechanics.

The introduction to the fundamental theory of TO could be conducted by using scientific literature about SO, such as "An Introduction to Structural Optimization" [24] and "Topology Optimization: Theory, Methods and Applications" [2]. In addition to these, the utilization of basic scripts, such as the 99-line script for TO by Sigmund [28] written in Matlab or the equivalent 200-line Python script, which can be used in open-source software, could support the theory presented in Section 3 with a numerical implementation. The different sections of the code could be presented, and relevant optimization exercises for simple structures could be given. The script's flexibility affords the opportunity to students to practice with the code and the essential equations of TO by changing the geometry, the boundary conditions, the load cases, and the material with respect to the models given in the exercises. Furthermore, they are challenged to confront the checkerboard problem and understand the need for continuous variables in the solution of the TO problem, as well as the reduction of the intermediate elements, by using the SIMP method with different penalization factors. An example of a more advanced exercise is the presentation of different optimization filters, such as sensitivity filtering [29,30] and black-and-white (Heaviside) filtering [31], which are presented in the updated 88-line Matlab script by Andreassen, et al. [32]. Finally, the new generation of the 99-line Matlab code for compliance

topology optimization and its extension to 3D by Ferrari and Sigmund [33] could be used in more advanced exercises. A TO example of a cantilever beam using the aforementioned Matlab scripts is depicted in Figure 5.

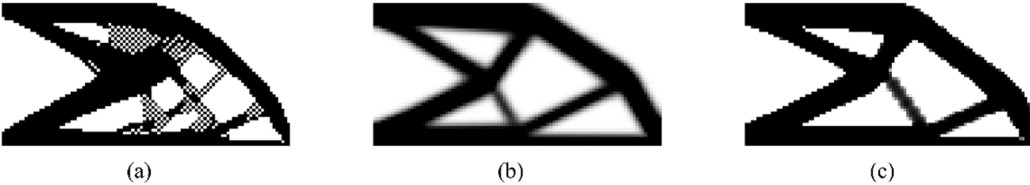

| (a) | (b) | (c) |

**Figure 5.** Topology optimization of a cantilever beam with a mesh size of 100 × 50 (horizontal × vertical) using Matlab scripts [28,32]: (**a**) checkerboard problem, (**b**) sensitivity filter, and (**c**) Heaviside filter.

In this exercise, students can optimize simple structures by using these codes and can try to identify differences in their optimized results. In addition, they can import the derived optimized designs into CAD software, such as SolidWorks or Fusion360, where they can use them as a canvas for the 3D modeling and the FEMs of the results, which would contribute to the learning of different CAD and FEM tools. The most popular commercial software for topology optimization is still based on the core 99-line Matlab code [28]. Fusion360 may be preferred due to the fact that it contains a free student license that includes a topology optimization module. Different exercises can be conducted in a CAD environment, where students can practice using tools for TO, FEM, and CAD through the design of the initial design concepts and the redesign of the optimized design solutions. In addition, postgraduate students can explore the impact of the designers' input on the optimized results, as by Tyflopoulos and Steinert [34].

An example of an optimization procedure is presented in Figure 6, where the students were asked to optimize a cantilever beam for its mass based on the given boundary conditions and load cases.

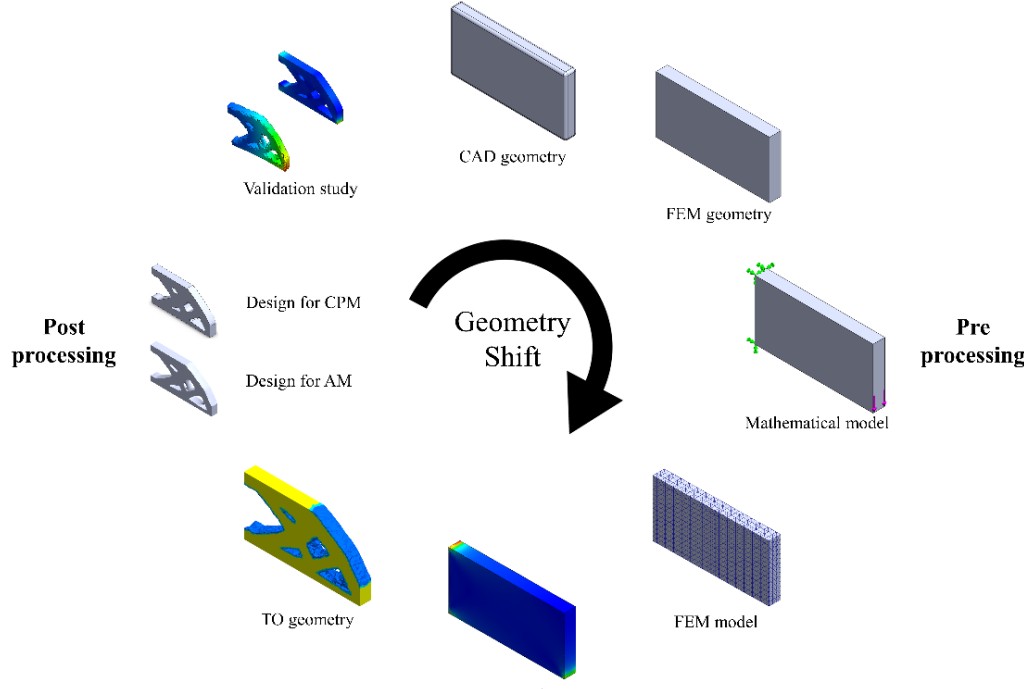

**Figure 6.** The TO-procedure of a cantilever beam based on Tyflopoulos, et al. [35].

On the one hand, the students practice both the pre-processing and the post-processing phases of the optimization procedure. In the pre-processing phase, they have to identify the boundary conditions and load cases of the given structures, create mathematical models, and conduct the required FEMs. In the post-processing phase, they redesign the optimized solutions for X, where X is either AM or CMP. For AM, the design solutions should be prepared for 3D printing, while for CPM, additional parameters should be added to support the manufacturability by conventional means. These two options will introduce the fundamentals of 3D printing and the traditional production methods, respectively. Finally, numerical validation studies using FEMs should be implemented at the last step in the optimization methodology presented in Figure 6. The production of the design solutions, as well as their experimental validation, could support the numerical analysis and offer students a complete product development process. A qualitative and quantitative comparison between the numerical and the experimental results could help students to identify the challenges and limitations of TO and improve their skills.

On the other hand, as a more advanced exercise, they could explore these limitations and challenges. For example, Figure 7a presents the impact of the design space utilized in TO with a desk as an example. From this example, the students could understand that by increasing the design space, the flexibility of the optimized algorithm is also increased, leading to better-optimized solutions. Another example of TO could be the integration of design-of-experiment (DOE) practices in its methodology and the presentation of its sensitivity to the given parameters. Figure 7b presents the sensitivity of the algorithm to changes in the boundary conditions. Small failures in a designer's inputs can easily lead to a lack of feasible design solutions. It is crucial for students to understand that TO can be used for both design inspiration and manufacturing by using the appropriate inputs.

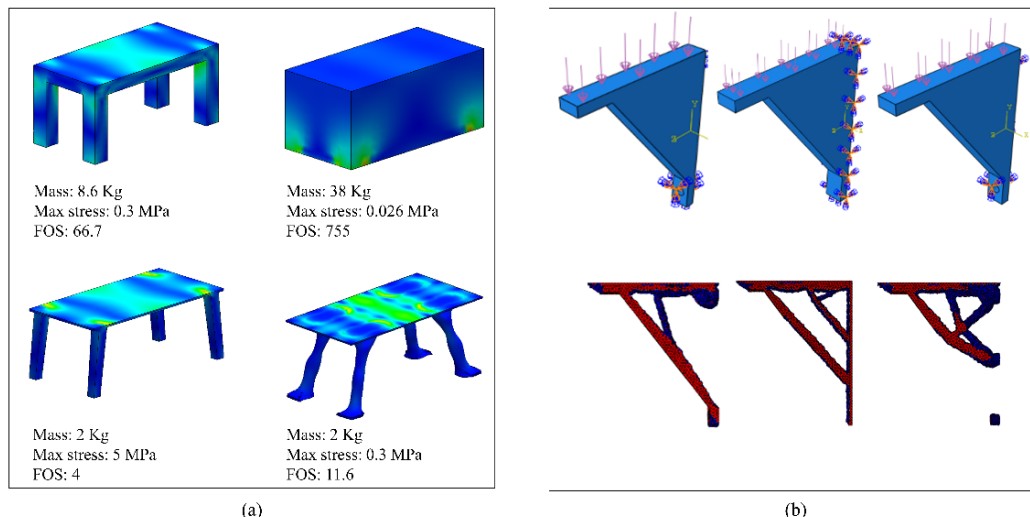

**Figure 7.** Examples of advanced TO exercises: (**a**) TO of a desk: impact of the utilized design space [36]; (**b**) TO of a wall bracket: the sensitivity to the given boundary conditions; adapted from [34].

In addition, projects that use real-life products in cooperation with industry could be used for the in-depth understanding of TO. In this way, students can apply their knowledge to real projects and optimize existing products for both their own benefits and the companies' sake. Some examples are the optimization of a ski binding [35] and the optimization of the brake calipers of a student racecar [37], as shown in Figure 8.

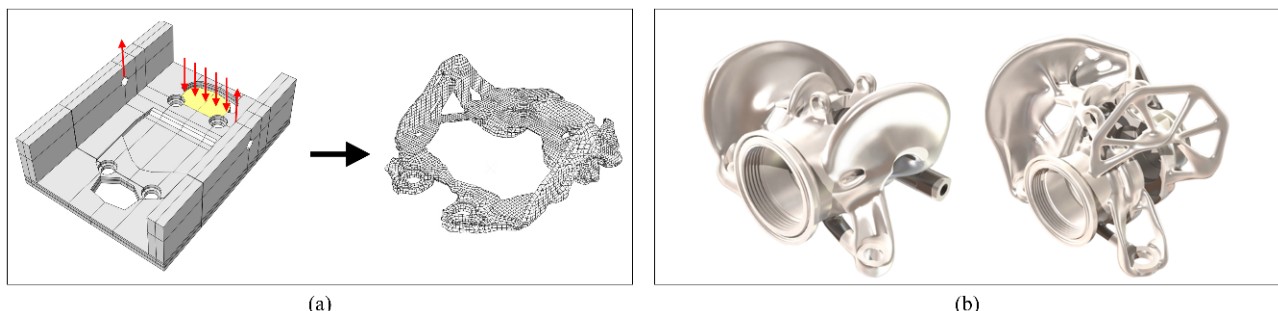

**Figure 8.** Examples of TO projects conducted by undergraduate students: (**a**) optimization of a ski binding [35] and (**b**) optimization of brake calipers [37].

## 5. A Teaching and Learning Framework for TOBL

In this section, a teaching and learning framework for topology-optimization-based learning (TOBL) is presented under the prism of the CDIO approach. The goal of this framework will be to educate under- and postgraduate engineering students in order for them to acquire the skills of modern CAD designers. This TOBL framework is a collection of useful theoretical fundamentals, as well as examples of open-access TO tools, exercises, and assignments, thus enabling its adoption in any study program in CAD engineering. The process used here follows the guidelines provided by the CDIO Initiative for the development of both course and integrated curricula, as presented in Section 3. For the identification of the needs and goals of TOBL, the process begins with the description of the attributes expected of a contemporary CAD designer.

### 5.1. The Goals of a Study Program in CAD Design Based on the Necessary Attributes of a Contemporary CAD Designer

Modern design is challenging because of the multitude of characteristics expected in newly developed products, which should be robust, attractive, and environmentally friendly. The continuous integration of new technologies in product design development has generated an increasing demand for different skill sets. Hence, the responsibilities and, consequently, the needs of CAD designers have dramatically changed. It is important to understand that a contemporary designer should not only design, but also engineer. In this section, the needs of a CAD designer are identified based on his/her contribution to the product development process. The product development process presented in this paper follows the model of Eppinger and Ulrich [38]. According to this model, the product development process is divided into six phases: planning, concept development, system-level design, detail design, testing and refinement, and production ramp-up.

During the planning phase, the designers should investigate the market and the technological changes and should generate new product ideas with respect to both the industry's strategies and societal needs. In addition, they should be able to identify the key stakeholders who are interested in a product, understand their needs, and translate them into product specifications, in addition to making business and project plans. These tasks increase the need for designers' skills to include problem solving, innovation, project management, and entrepreneurship.

The second phase is the conceptualization of the dominant product ideas. At this point, the designers create several design concepts based on the stakeholders' needs and by utilizing different design tools and algorithms. The CAD software is the most commonly implemented design tool because it increases designers' productivity with the flexibility of their design [39]. Furthermore, the designers are able to both create and test experimental prototypes. These prototypes can be both numerical and physical. On the one hand, numerical prototypes are CAD designs. These designs can be tested and validated using FEMs. On the other hand, physical experimental prototypes can be developed with



inexpensive materials, such as cardboard and MDF. Hence, a CAD designer should have good design, computer, and building skills.

The system-level design addresses the product's architecture, as well as its decomposition into subsystems and components. In addition, the product's boundaries and relationship with the operating environment should be defined. This phase helps designers to create a clear overview of the end product without building or developing a fully detailed system design. However, in this phase, many important decisions are made, such as the identification of both the functional and physical elements of a product, the way that the elements will be combined to make an assembly, and possible manufacturing methods for each of them. These activities demand both design and engineering skills, such as knowledge and simulation of both the additive and conventional production methods that use CAM tools and knowledge of mechanics and dynamics, to mention a few.

In the detail design phase, CAD designers should develop technical drawings that include tolerances and materials for each of the product's components and "translate" these into machine movements by using software coding. A detailed list of all components should be created, including all in-house and purchased parts, as well as their numbers and descriptions. Furthermore, the Design for Excellence (DfX) parameter should be taken into account, where X can represent the manufacturability, cost, robustness, and other traits or features [40]. High design and engineering skills are also demanded here, such as for material choices and production methods.

The testing and refinement phase involves the production and evaluation of multiple preliminary versions of the product. Here, the CAD designers test and compare their designs with the numerical ones and make possible changes and modifications. Moreover, additional optimizations, such as TO, can be conducted with a focus, for example, on the reduction of the weight or cost of the product.

Finally, the final design solutions are gradually transformed into new realities in the production ramp-up phase. The CAD designers should monitor the manufacturing of the products and should be prepared to insert last-minute changes. In addition, the collection of feedback from the key stakeholders can help to identify possible product errors and omissions in early qualification of the production. The modification of the designs based on this feedback can lead to possible product improvements.

Ultimately, designers work in teams, which means that they communicate and collaborate with other engineers during the entire product lifecycle. Thus, there is a great need—as for any engineer—for communication and teamwork skills that support disciplinary cooperation and design activities. It is clear that the boundaries and the skills described are relevant in all product development phases. Hence, in order to list the necessary attributes of a CAD designer, the product development process should be seen holistically.

To conclude, a designer should understand and implement the whole product development process by using different tools (drafts and CAD), making concepts, building prototypes, testing, validating, and optimizing (SO). This is akin to the CDIO's "conceive, design, implement, and operate" activities. Hence, a study program that adapts the CDIO Syllabus can educate a student to become an effective modern CAD designer. A presentation of the needs of a CAD designer that combines the product development process of Eppinger and Ulrich [38] and the CDIO approach is depicted in Table 1.

The CDIO approach has the goal of educating students to be able to develop new products, processes, and systems by utilizing the deeper working knowledge of technical fundamentals that they acquire and by considering the impacts of research and technology on society [19]. Thus, TOBL must offer a deep working knowledge by using active learning TO tools that can lead to conceptual understanding of CAD design [41]. Students following a TOBL study program utilize their knowledge to design and develop optimized CAD products that integrate technological changes and satisfy societal needs. In addition, the material savings provided by the TO could support the production of products that minimize raw material resources. Until recently, traditional design placed both the refinement and optimization of the products at the latter phases of product development (phase five

in the model of Eppinger and Ulrich). According to our proposed framework, designers could avoid design fixation and offer optimized designs from the beginning, thus saving valuable time and money.

**Table 1.** The attributes of a contemporary CAD designer.

| | | |
|---|---|---|
| Conceive | Planning | Problem solving and innovation<br>Project management and entrepreneurship |
| | Concept Development | Good design, computer, and building skills<br>Creativity |
| Design | System-Level Design | CAD and CAM<br>Engineering fundamentals |
| | Detail Design | Design and engineering skills<br>Material choice<br>Production methods |
| Implement | Testing and Refinement | FEM and Optimization<br>Physical testing |
| Operate | Production Ramp-up | Monitoring the manufacturing<br>Logistics<br>Maintenance |

*5.2. CDIO Syllabus for a CAD Designer*

As mentioned in Section 1, the CDIO Syllabus can be adapted to any CAD design study program based on its distinct needs. In particular, it is mainly the first part of the syllabus—the collection of the disciplinary knowledge and reasoning—that can deviate, while the other three—the personal and interpersonal skills and attributes, as well as the CDIO philosophy—are required, for the most part, from all engineering graduates regardless of their study program.

The first part of the syllabus is divided into three categories at its second level of detailed content—i. knowledge of underlying mathematics and sciences, ii. core engineering fundamental knowledge, and iii. advanced engineering fundamental knowledge, methods, and tools. These are further elaborated upon in the following.

i.  Any engineer, including a CAD designer, requires mathematics as underlying knowledge in his/her education. Specifically, a designer following TOBL should be familiar with algebra, calculus, analysis, and, indisputably, geometry and topology, as these can be considered as prerequisites for CAD and TO. In addition, dynamic systems with differential equations and mathematical physics with a focus on classical mechanics support the FEM courses. Furthermore, the theory of applied statistics provides fundamentals for parametric and non-parametric statistical models while introducing DOE to the students. Basic physics and chemistry with a focus on classical mechanics and stereochemistry support the core engineering fundamental knowledge. Finally, basic knowledge of programming languages and computer programing will help new designers to develop their own scripts and understand the different TO algorithms.

ii. The core fundamental engineering knowledge is almost the same in any engineering undergraduate study program. However, the focus should be adapted to the students' needs. Concerning the studies of a CAD designer, mechanics, dynamics, thermodynamics, material science, and structural analysis are important for the understanding of basic engineering concepts and can be used in the design parametrization of CAD models, the implementation of FEM simulations and validations, the material selection, and the solutions of optimization problems. In addition, knowledge of conventional manufacturing processes (CMPs) and additive manufacturing (AM) will be utilized in the testing and refinement step of product development, and both should be accounted for in the optimization process.

iii.   A special focus on the 3D printing, CAD, FEM, CAM, and TO methods and tools constitutes the advanced engineering fundamental knowledge, methods, and tools. In addition, statistical and computer programming software are demanded, among other things. All of these will help students to apply the theory learned, to develop projects, and to learn through application. The active learning tools, algorithms, and assignments presented in Section 4 will make significant contributions to this section of the syllabus.

The other three parts contain the more generic knowledge, skills, and attributes that all engineering graduates should possess, and they are described thoroughly in the CDIO Syllabus. In general, a CAD designer should be able to identify, understand, model, and solve any kind of optimization problem by utilizing different optimization methods and approaches while working either alone or in teams and communicating and cooperating efficiently with other team members. Thus, good personal, professional, and communication skills are demanded. Contemporary CAD designers should communicate continuously with other engineers, such as industrial and AM engineers, during a product's lifetime. Good communication can lead to effective product development and reduces the uncertainty among its phases, resulting in better products and minimizing the need for rework and waste. Finally, the syllabus should be in accordance with the philosophy of CDIO and TOBL and, thus, educate students to function in and contribute to an enterprise in a societal and environmental context.

*5.3. Course Design and Integrated Curriculum in TOBL*

The development of both TO courses and an integrated TOBL curriculum is a demanding procedure, and these should be seen together. On the one hand, an integrated TOBL curriculum should be developed by the corresponding faculty with respect to the necessary skills of a contemporary CAD designer and the participation of all of the related key stakeholders. The courses involved should cover the disciplinary knowledge mentioned in Abstract. Of course, this knowledge can vary from program to program based on the students' specializations. A balance between mandatory courses and a plethora of elective courses can offer several topics that can cover every designer's future needs for knowledge and skills. On the other hand, the planning of a TO course should begin with the definition of its learning outcomes, followed by the teaching and learning activities, as well as the course's assessments. The learning outcomes of a TO course should be seen as a part of an integrated curriculum and not independently. In addition, they can be categorized with respect to their difficulty and level of understanding. The teaching and learning activities of a TO course should be aligned with the tools presented in Section 4. These active learning tools, together with the implementation of real-life optimization problems, will promote the practical application of TO. Finally, suitable assessments could both measure students' understanding of TO and evaluate the individual courses, as well as the integrated TOBL curriculum.

An example of a teaching and learning framework for TOBL based on the findings of this research work is presented in Table 2. This framework could be used in the process of the development of an integrated curriculum in any TOBL study program in the scope of five academic years. Table 2 recommends the essential knowledge that can support TOBL, examples of teaching and learning activities, and intended learning outcomes and assignments. In addition, the learning outcomes are classified with respect to the academic year and the Feisel–Shmitz taxonomy.

**Table 2.** Example of a teaching and learning framework for topology-optimization-based learning (TOBL).

| Teaching and learning framework for TOBL | | | | |
|---|---|---|---|---|
| Underlying/Essential Knowledge that Can Support TOBL | | | | |
| Mathematics Physics | Mechanics Dynamics | Thermodynamics Programing | Material Science Statistics | Chemistry |
| Bachelor: Core Eng. Knowledge | | | Master: Advanced Eng. Knowledge | |
| Teaching and learning activities | | | | |
| 1st Year | 2nd Year | 3rd Year | 4th Year | 5th year |
| Figures/Examples with initial and optimized designs Interactive games and apps Matlab/Python exercises: 99-line script CAD/FEM exercises Small group projects | Matlab/Python exercises: different scripts CAD/FEM exercises Applications in structural problems Small group projects | Bachelor dissertations in groups: Optimization of real products in cooperation with industry | Theory and examples of different SO methods and algorithms Exercises combining SO with DOE and sensitivity analysis Applications in structural and multi-physics problems Small group projects | Individual Master thesis: Design and optimization of real products in cooperation with industry |
| Intended Learning Outcomes | | | | |
| Excite curiosity and increase motivation Introduction to programming languages Mechanical design Introduction to 3D modeling Introduction to FEM Introduction to CMP and AM | TO script TO challenges TO for AM vs. TO for CPM Moderate CAD Moderate FEM Parametric design Statistical analysis Product development Reverse engineering Design thinking | In-depth understanding of TO Create, analyze, and evaluate different optimization problems Plan, prepare, lead, and manage projects Contribute to research and development work | SO scripts and software Advanced CAD Advanced FEM DOE Sensitivity analysis Statistical analysis | In-depth understanding of SO Create, analyze, and evaluate different optimization problems Plan, prepare, lead, and manage projects Contribute to research and development work |
| Assessment | | | | |
| Exercises/Exams Small group projects | Exercises/Exams Small group projects | Group Bachelor dissertations | Exercises/Exams Small group projects | Individual Master dissertations |
| Level in Feisel–Shmitz taxonomy | | | | |
| Define | Compute | Explain | Solve | Judge |

## 6. Discussion

The TOBL framework developed here includes all of the educational aspects of TO. Hence, to answer the RQs posed in the beginning of this paper, it can be stated that:

- TO is a useful multi-educational tool that can be effectively utilized to introduce and teach the different educational elements that constitute a degree in CAD engineering, such as CAD, FEM, CAM, AM, and CPM. In addition, its application to real-life products can offer theoretical insights to the students about product development, design thinking, and reverse engineering.
- There are a plethora of open-source active learning tools concerning TO that could easily facilitate both its introduction and in-depth understanding among under- and postgraduate students. A study program that supports TOBL under the prism of the CDIO initiative offers a learning and educational method that can create contemporary CAD designers who can design and develop optimized products aligned with technological changes and societal needs.
- The prerequisite knowledge that is demanded and that can support TOBL consists of the basic fundamental engineering knowledge that is included in any undergraduate

engineering program. However, additional focus on special topics related to TOBL should be covered, such as topology, mathematical physics, classical mechanics, computer programing, and applied statistics.

- TO can be taught to both undergraduate and postgraduate students studying CAD engineering from their first academic year. However, theoretical topics, exercises, applications, and projects with a gradually increasing difficulty can be used during the different academic years, leading to increased levels of understanding of TO.

There are just a few previous works in the literature that either promote TO as a teaching tool in engineering [42–44] or implement the CDIO Syllabus in the development of engineering courses [45–47]. On the one hand, de Oliveira, Steffen, de Moraes Vasconcellos and Sanchez [42] and Mullins, Kirkegaard, Jessen and Klitgaard [43] proposed practical and simplified models based on TO that could be easily integrated into the undergraduate architecture, while Sangree, Carstensen, Gaynor, Zhu and Guest [44] explored the potential role of TO as a teaching tool in structural engineering education. On the other hand, Quist, Bhadani, Bengtsson, Evertsson, Malmqvist, Enelund and Hoffenson [45] developed an engineering design and optimization course based on the CDIO Standards. In addition, Deweck, Kim, Graff, Nadir and Bell [46] presented an undergraduate design and rapid prototyping course in the Department of Aeronautics and Astronautics at MIT that combined CAD/CAE/CAM and CDIO. Finally, in their research paper, Zhong, Chiu and Lai [47] measured students' cognitive load and flow experience by using CDIO engineering in a flipped programming course.

However, the research work conducted in this paper was the first attempt to bridge the educational benefits of TO and the CDIO Syllabus, resulting in a novel educational framework, the TOBL framework, which can be easily applied in both curriculum and course development in any program of CAD engineering study.

## 7. Conclusions

The possibility of using TO as an educational tool for CAD, FEM, CAM, AM, and CPM for under- and postgraduate students in CAD engineering studies is explored here. TO is shown to be adaptable and relevant in all academic years, either as an individual course or as an integrated curriculum. The CDIO Syllabus, together with the implementation of open-source active learning tools about TO, such as figures, interactive exercises, and games, offers the students different levels of understanding of TO. The findings in this paper resulted in a novel learning and teaching framework for topology-optimization-based learning, the TOBL framework. The underlying knowledge that can support TOBL, its teaching and learning activities, and its intended learning outcomes was presented in detail. The TOBL framework can educate contemporary CAD designers who can conceive, design, implement, and operate optimized products. With this approach, traditional CAD design, where the refinement of the developed products was one of its last phases, is now replaced with a design methodology that is oriented towards optimization. In this way, a contemporary designer avoids the design fixation that leads to known CAD geometries and is able to explore new, lighter, and more robust design ideas while saving material and development time. Finally, the authors encourage the TO community to exchange ideas, knowledge, and experience that can contribute to TOBL in any study program that involves CAD engineering.

**Author Contributions:** Conceptualization, methodology, software, writing—original draft preparation, E.T.; writing—review and editing, E.T. and C.H.; supervision, C.H. and M.S. All authors have read and agreed to the published version of the manuscript.

**Funding:** This research received no external funding.

**Informed Consent Statement:** Not applicable.

**Data Availability Statement:** Not applicable.

**Conflicts of Interest:** The authors declare no conflict of interest.

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
