# Peer review of "Topology-Optimization-Based Learning: A Powerful Teaching and Learning Framework under the Prism of the CDIO Approach"

_education, doi:10.3390/educsci11070348_

Round 1
Reviewer 1 Report
Good afternoon,
I congratulate the authors. My opinion is that the paper is published with little improvements:
1) Page 6, lines 218-221, there are spelling mistakes.
2) An important issue for the reader is to know the educational level that the paper is directed to. But until page 15 it doesn´t appear. I think it is important, so it has to appear in the tittle or abstract. It seems it is for university studies (Engineering). But I think it is important to say it from the begining.
Reviewer 2 Report
The present paper is dealing with a very interesting subject studied only by a few authors.
I appreciate the paper, but there are some problems to be explained.
The acronym CDIO is not explain until line 104, and maybe it is to late …
The usual sections Methodology (maybe section 3) – Methods and Results (maybe section 4 and 5) are not clearly defined.
Anyway, it is not clear how section 3 is applied to the following sections.
Is the optimization theory necessary here?
If yes, you must demonstrate the application in the next sections.
Discussion
This section is missing.
Other works must be compared in this section.
Maybe the answers to the RQ (line 571-591) could be analyzed in a discussion section and compared with previous woks.
Conclusions
The conclusions must be more detailed.
Minor corrections
Correct line 218 – 221 Error! Reference source not found.
Round 2
Reviewer 2 Report
Congratulations to the authors!